# Polyvalent Snake Antivenoms: Production Strategy and Their Therapeutic Benefits

**DOI:** 10.3390/toxins15090517

**Published:** 2023-08-24

**Authors:** Kavi Ratanabanangkoon

**Affiliations:** Department of Microbiology, Faculty of Science, Mahidol University, Bangkok 10400, Thailand; kavi.rtn@mahidol.ac.th; Tel.: +66-86-974-8874

**Keywords:** antivenom, snake, syndromic polyvalent, polyspecific, equine, neurotoxins, hematotoxins

## Abstract

Snake envenomation remains an important yet neglected medical problem in many countries, with around five million people affected, and over a hundred thousand deaths annually. Plasma-derived antivenoms are the main therapeutic agent available. Monovalent antivenoms are produced via the immunization of large animals, e.g., horses, with one venom, after which the horse serum can neutralize the homologous venom, with minimal or no cross neutralization against other venoms. It is necessary, therefore, for the culprit snake to be identified, so that the appropriate specific antivenom can be selected. Polyvalent antivenoms (pAVs) are produced via immunization with a number of snake venoms, and the serum can neutralize all the venoms used in its production. Thus, pAVs can be used to treat several venoms from a country/region, and the identification of the culprit snake is not necessary. There are various parameters and processes involved in the production of pAVs, depending on the requirements and resources available. Most commercial pAVs use a mixture of both elapid and viperid venoms as immunogens, while some pAVs use either elapid or viperid venoms. Some pAVs are produced through the mixing of more than one monovalent or polyvalent antivenom. These various types of pAVs have their own characteristics, and have benefits and drawbacks. The major benefits of pAVs are the wide coverage of many medically important venoms, including many heterologous venoms. They also remove the need to identify the culprit snake, and they can be produced at a lower cost than several monovalent antivenoms. Interesting polyvalent antivenoms, termed ‘syndromic pAVs’ (s-pAVs), have recently gained attention. They are produced for use according to the syndromes manifested in snakebite patients. The venoms that produce these syndromes are used as immunogens in the production of ‘syndromic antivenoms’. For example, ‘neurotoxic polyvalent antivenom’ and ‘hematotoxic polyvalent antivenom’ are produced using the neurotoxic elapid and hematotoxic viperid venoms as immunogens, respectively. They were first marketed by the Thai Red Cross in 2012, and have since gained attention as a possible therapeutic modality to help solve the problem of snakebite envenomation globally. The merits of these s-pAVs, including their efficacy and wide paraspecificities, are discussed.

## 1. Introduction

Snakebite envenomation remains a seriously neglected tropical disease. It has been estimated that at least 1.8–2.7 million people are affected annually, with about 94,000–125,000 deaths [1,2]. Successful treatment requires the timely administration of an effective antivenom (AV). In the present article, animal-plasma-derived antivenoms will be discussed.

Antivenoms are produced via the immunization of large animals, such as horses, donkeys, sheep, or camels, using the crude venom(s) or toxin(s) from the relevant snake. The plasma from these animals can be prepared, with the red blood cells being returned to the source animals. The plasma is then fractionated to obtain the immunoglobulins, which may be further treated with proteolytic enzymes to give Fab or F(ab’)_2_ antibodies.

Originally, antivenom was produced via the immunization of an animal with a single venom, and the antiserum obtained was used to treat patients envenomed by the cognate snake venom used in its production. This type of AV is called a monovalent AV, is usually effective only against that particular venom, and often shows little or no cross-neutralization with other heterologous venoms [3]. Therefore, it is necessary for the species of the culprit snake to be correctly identified, in order to select the specific antivenom for effective treatment.

However, correct identification of the culprit snake is frequently difficult, as snakebites usually take place in the dark, or in bushy areas. In Thailand, culprit snakes are identified in about 44% of envenoming cases [4]. In addition, the development and use of snake diagnostic test kits are expensive. Such tests may require about half an hour to provide a result, and this will delay the appropriate antivenom administration. Further, the half-life of these diagnostic tests is usually rather short, and this makes the test quite expensive. For these reasons, it is advantageous to produce polyvalent antivenoms (pAVs) that can effectively neutralize a spectrum of snake venoms that are medically important in a country/region, without the need to identify the culprit snakes.

Polyvalent or polyspecific antivenoms are produced with the purpose of neutralizing multiple snake venoms and, thus, can provide effective therapy against a selection of the medically most important snakes within a geographical area. It is expected that the widespread adoption of pAVs will negate the necessity of identifying culprit snakes. A single pAV, instead of a battery of different monovalent antivenoms, would, thus, suffice for snakebite treatment in each country or region. It should be mentioned here, however, that, after the administration of the appropriate antivenom, it is useful to know, or to deduce, the likely causative species, so as to predict the clinical course, and to provide the optimal supportive treatment and clinical management.

## 2. Various Parameters Involved in the Production of Polyvalent AVs and the pAV Characteristics

The first pAV was produced by Vital Brazil in 1898, at the Instituto Butantan, São Paulo, Brazil [5]. Since then, a variety of procedures have been employed in pAV production, resulting in differing therapeutic characteristics [6,7,8,9]. The following section details various procedures used in pAV production, and the properties of the resulting pAVs.

### 2.1. The Number of Venoms to Be Included in the Production of a pAV

Conventional pAVs are prepared via the immunization of horses with a mixture of various crude venoms. Naturally, producers want to include as many immunizing venoms as possible, so that all the target venomous snakes are covered. As each snake venom can contain more than 100 different proteins [10], immunization with a mixture of more than 5–6 venoms may result in a total protein antigen load that can overwhelm the immunized animal, and result in a low titer of the produced antibodies. Even after fractionation and concentration, the final antivenom products might be ineffective against some of the homologous venoms. This could possibly be due to the effect of ‘antigenic competition’ [11] resulting in lower potencies in the AV produced. Therefore, there are certain upper limits to the number of snake venoms that can be used in the immunization of horses. Usually about 5 or 6 are used [9].

### 2.2. Use of Crude Venoms or Toxin Fractions

One way to circumvent the perceived ‘antigenic competition’ problem is to first fractionate the venoms, to remove highly immunogenic but non-toxic proteins from the venoms, and then use the separated lethal toxin fractions for immunization [12]. In this way, the amounts of immunizing toxin proteins are greatly reduced, allowing an increase in the number of venoms used in the pAV production, up to a total of 12 different venoms. As the horse B cell antibody repertoire is almost infinite, antibody paratopes against all the toxin epitopes could be generated [13]. This should significantly increase the paraspecificity of the pAVs produced. It has been shown that this approach could lead to pAVs with an improved potency [14] and a wider paraspecificity [12,15]. However, this approach is applicable only in cases where all the toxins lethal to humans are known, and can be fractionated without too many hurdles.

### 2.3. Antagonism and Synergism between the Constituent Venoms Used in pAV Production

It has been demonstrated that the *Crotalus simus* and *Crotalus durissus ruruima* venoms exert a deleterious effect on the antibody response towards *Bothrops asper* venom when used as co-immunogens in pAV production [8]. This study is interesting and important, and it raised awareness regarding the potential for positive or negative interactions among venoms in pAV production. Interestingly, it has also been shown that a mixture of specific venoms can stimulate the antibody response to a greater extent than would immunization with only a single venom in the production of bothropic–crotalic pAVs [8]. In this case, it seems that various venoms may act as ‘adjuvants’, to enhance the immunogenicity of another venom. This study could have significant implications in pAV production, and must be considered in any pAV production plan. More research is also needed, to shed light on the mechanisms underlying these interactions.

### 2.4. Production of ‘Mixed pAVs’

Another approach to overcoming the problem of too many immunizing venoms in pAV production is to produce 2 or more pAVs, each using fewer immunizing venoms, and then mix the IgG or F(ab’)_2_ of these pAVs, to obtain a ‘mixed pAV’. This approach has been successfully employed via the mixing of the F(ab’)_2_ of one pAV covering five Bitis and Echis venoms with the F(ab’)_2_ of another pAV covering six elapid venoms, to give a ‘mixed pAV’, which could neutralize 11 snake venoms. This ‘mixed pAV’ was shown to have a wide paraspecificity against the venoms of numerous snakes in sub-Saharan African countries [9]. The mixing of monovalent antivenoms to give a pAV has also been tried [16]. However, the drawback to this approach is the increased expense, due to the requirement for more horses, the additional cost of fractionation, etc. Another concern linked to this approach is the possible mutual dilution of the constituent antibody pools. The approach can provide a wider paraspecificity, but may reduce the potency per unit volume of the resulting ‘mixed pAV’.

Victims of snakebites most likely suffer from either elapid or viperid envenoming. When they are treated with a ‘mixed pAV’ that contains both anti-elapid and anti-viperid antibodies, one of the two antibody populations is not used, and is, therefore, wasted. In addition, the unused antibody immunoglobulins may themselves contribute to adverse reactions. Further, a ‘mixed pAV’, such as that described by Ramos-Cerrillo et al. [9], that showed a good potency and paraspecificity would, without the mixing of the two groups of F(ab’)_2_ antibodies, serve as two independent syndromic pAVs, one for use against elapid envenoming, and the other against viperid envenoming (see below). Finally, in the same total quantity, these two syndromic pAVs could be used to treat twice as many snakebite victims as the ‘mixed pAV’.

### 2.5. pAVs with Combined Anti-Elapid and Anti-Viperid Activities

Many pAVs are produced using a mixture of elapid and viperid venoms. These ‘combined pAVs’ (c-pAVs), which offer both anti-elapid and anti-viperid activities, have been produced, and are widely used in many countries [14,17,18,19], where they have saved countless lives. However, there are some possible drawbacks associated with this type of pAV.

The major commercial pAVs produced in India are against the four most medically important snakes, “The Big 4”, consisting of two elapid and two viperid venoms (*Naja naja*, *Bungarus caeruleus*, *Daboia russellii*, and *Echis carinatus*). It was apparently assumed that all the venom toxins were immunologically equivalent. However, this is not the case. The lethal toxins of these two types of snake are vastly different, not only chemically, but also immunochemically. Almost all elapid venoms contain neurotoxins and cytotoxins of the 3-finger toxin (3FTx) family, with molecular weights of around 7 kilodaltons [20,21], and are poorly immunogenic [22,23]. In contrast, the viper venoms mostly contain high MW toxins of around 20–30 kilodaltons, with enzymatic activities, e.g., phospholipases A_2_, zinc-dependent metalloproteinases, and serine proteinases; these hematotoxins are quite immunogenic. Therefore, it is not surprising that these pAVs contain more antibodies against the viperid protein toxins than against the elapid toxins (Figure 1) [23].

Various immunochemical studies have shown that more antibodies in the c-pAVs recognize and neutralize viper venom proteins, while the elapid toxins are not well recognized [24]. Thus, these pAVs often show higher neutralizing potencies against the homologous viper venoms, but usually a poor or even absent neutralization against the elapid venoms [25]. Put another way, the neutralizing potencies of the anti-viper activity in the c-pAVs are often higher than those of the anti-elapid activity [26]. Thus, the producers’ recommended initial doses of c-pAV for the treatment of elapid and viperid envenoming may be different. But, as the culprit snake is usually not identified, choosing the appropriate initial dose of c-pAV for treatment may be challenging.

### 2.6. Paraspecificity of pAVs

It has been widely observed that various pAVs may provide cross-neutralization against homologous venoms with significant geographical variations in venom profiles [27], or against other heterologous venoms, even from different genera [19,28,29,30,31,32]. This assumed paraspecificity may be useful in providing a wide coverage against other venoms in places where no specific AV is available. The paraspecificity of pAVs is more pronounced than that observed in monovalent AVs. A possible basis for this is that the horse, when exposed to the ‘diverse toxin repertoire’ present in a mixture of many immunizing venoms [12], could respond by producing the corresponding diverse antibodies capable of reacting with similar epitopes in the toxin isoforms of other related venoms [15,27]. An example is the neutralization of *Hypnale hypnale* venom from Sri Lanka by Hemato Polyvalent AV, produced in Thailand [33,34]. In Central American and European countries, and elsewhere, pAVs against various vipers have been produced, and they have shown a wide paraspecificity against numerous other vipers in the region [19,29,35,36,37,38].

### 2.7. The Benefits and Advantages of pAVs

pAVs are produced against the venoms of many snakes that are medically important, and they could save lives, especially in areas where no specific antivenoms are available.

pAVs could be used to treat envenoming by unidentified snakes in specified countries/regions. As most snakebites occur in the dark, or in bushy areas, the culprit snakes are rarely identified. The use of specific monovalent AVs may be ineffective, cost lives, and lead to the wastage of expensive antivenoms. In contrast, with regard to the selection of the appropriate antivenom for treatment, pAVs remove the need to identify the culprit snake, and the use of snake-identifying test kits, which are expensive and time consuming.

pAVs can be administered immediately, as there is no need to identify the culprit snake. The rapid administration of an effective pAV could reduce the severity of envenoming, and save lives.

Another advantage of pAVs is the lower cost of production, in that fewer groups of horses are needed, compared to the number required to produce several monovalent antivenoms. Moreover, the cost of the fractionation processes for the plasma samples, and the quality control of the products, packaging, inventory, etc., can be reduced, and this can make the final products less expensive and more affordable.

pAVs can replace some monovalent AVs against snakes that cause serious, but low, incidences of envenoming. For example, in Thailand, the *O. hannah* (*king cobra*) is a WHO Category 2 snake and, although it causes very few incidents of envenomation, at seven cases or 0.22% out of 3091 cases of venomous snakebites [4], a monovalent AV is produced. The inclusion of *O hannah* in the Thai pAV could eliminate the need to produce the monospecific anti-*O hannah* AV.

## 3. Syndromic Polyvalent Antivenoms (s-pAVs)

An interesting version of pAVs is called syndromic polyvalent antivenoms (s-pAVs). As the name implies, they are produced for the treatment of snake envenoming based on the signs and symptoms or syndromes manifested in the victims. Mixtures of venoms that can cause these syndromes are used as immunogens in s-pAV production.

In brief, the medically important venomous snakes causing most snakebite envenoming can be classified into two families, i.e., Elapidae and Viperidae [39,40]. The elapids produce mainly neurotoxins and cytotoxins of the 3FTx family, while the vipers produce hematotoxins. Thus, the syndromes that the snakebite victims present are used to determine which s-pAV treatment is appropriate. For example, victims showing signs of neurotoxic poisoning would be treated against neurotoxins, while those showing signs of hematologic disorder would be treated against hematotoxins. Specifically, patients with bleeding and coagulopathy (signs of hematologic disorder, Figure 2A) would be treated with a ‘syndromic hemato pAV’ produced using various viper venoms as immunogens. On the other hand, patients with muscle weakness, respiratory paralysis, or bilateral ptosis (signs of neurotoxin poisoning, Figure 2B) would be treated with a ‘syndromic neuro pAV’, produced using local elapid venoms as immunogens. The syndromic approach and syndromic antivenoms were first described by Williams et al. in 2011 [41].

The elapid and viperid venoms used in production are usually those of WHO Category 1 snakes, which are medically important snakes of the country or region. For example, in Thailand, two types of syndromic pAVs have been studied and produced, one against neurotoxic venoms [6], and the other against hematotoxic venoms [42,43]. These s-pAVs are called ‘Neuro Polyvalent Snake Antivenin’ and ‘Hemato Polyvalent Snake Antivenin’, and they have been produced and marketed by Queen Saovabha Memorial Institute of The Thai Red Cross Society since 2012 (Figure 3). The ‘neuro s-pAV’ is prepared using immunogens from three medically important elapid neurotoxic venoms of the country (*Naja kaouthia*, *Ophiophagus hanna*, and *Bungarus fasiatus)* [6] and, later, another elapid *(Bungarus candidus*) was added. This s-pAV is used in cases where snakebite victims show signs of neurotoxicity with muscle weakness, with bilateral palpebral ptosis (Figure 2B) [40]. The ‘hemato s-pAV’ is produced using three medically important hematotoxic venoms (*Daboia russellii*, *Calloselasma rhodostoma*, and *Trimeresulus albolabris*) as immunogens [42,43]; it is used when the patient shows signs of persistence of incoagulable blood and bleeding in the whole blood clotting test (20WBCT) for more than 20 min (Figure 2A) [44]. Thus, the syndromic approach allows the physician to observe the signs and symptoms of the patient to select the appropriate s-pAVs, and use the criteria (bilateral ptosis or 20WBCT) as indicators of systemic poisoning and the appropriate time to administer the antivenom. In these cases, the identification of the culprit snake is not needed.

It should be mentioned, at this point, that there are some medically important elapids, such as *Naja nigricollis* and *Naja ashei*, whose venoms do not lead to signs and symptoms of neurotoxicity. They do, however, give rise to a dominant syndrome of cytotoxicity. This type of venom could be considered as an additional immunizing venom in the production of ‘non-neurotoxic’ s-pAVs for the treatment of a snake envenoming that produces a clinical syndrome dominated by procoagulant, hemorrhagic, or cytotoxic effects.

### 3.1. The Paraspecificity of s-pAVs

Polyvalent antivenoms have been shown to cross-react with other heterologous venoms, and this cross-reaction is usually more pronounced than that observed with monovalent antivenoms. This paraspecificity is beneficial in the treatment of snake venoming, as it can provide a wider coverage of a larger number of snakes. This is likely to be the case with s-pAVs.

Each s-pAV is produced against venom toxins that give rise to similar signs, symptoms, and syndromes; i.e., they are raised against pharmacologically similar toxins. For example, in the case of the neurotoxic s-pAV, the venom immunogens are from neurotoxic elapids whose lethal components are the post-synaptic 3FTx family and, for some elapids, the phospholipase A_2_-containing presynaptic neurotoxin family. These toxins are chemically similar, and are structural homologs. Thus, the paratopes of the antibodies raised are likely to cross-react with the epitopes of the toxin isoforms, or the proteoforms of other elapid venoms [13]. This could lead to the wide paraspecificity observed in the neurotoxic s-pAV, and numerous published results attest to this conclusion [15,31,45,46,47].

Similar situations may prevail, or be present, for the viperid venoms. They contain various enzyme toxins, each with almost identical functions, e.g., serine protease, phospholipase A_2_, zinc-dependent metalloproteinases, hyaluronidase, etc. These enzymes from different vipers are likely to have some structural similarity, at least around the active sites of the enzymes. Antibodies raised against these enzymes have been shown to cross-react with the corresponding enzymes from other vipers [48,49]. The pAVs produced against several viper venoms have shown a very wide paraspecificity [30,50].

### 3.2. Possibility of Wider Paraspecificity in the Syndromic pAVs

The wide paraspecificity of the s-pAVs described above is inherent to their production process. However, it should be possible to further increase their paraspecificities, to widen the coverage of snakes in neighboring geographical areas. The number of immunizing venoms for each s-pAV is quite low, compared to those used in the ‘combined pAVs’, in which both elapid and viper venoms are used as immunogens. This will allow for an increase in the number of venom–immunogen cocktails in s-pAV production. Thus, it should be possible to produce an effective s-pAV covering a wide geographic area, such as various countries in Asia and in Africa.

For South Asia, the elapid venoms required to produce a ‘neuro s-pAV’ for use in seven countries in mainland South Asia may include a total of about seven WHO Category 1 elapid venoms: *Naja oxiana*, *Bungarus caeruleus*, *Bungarus niger*, *Bungarus walli*; *Naja kaouthia*, *Naja naja*, *Bungarus sindanus*, and possibly some other *Bungarus* species [51]. Similarly, a ‘hemato s-pAV’ may involve seven WHO Category 1 viper venoms: *Echis carinatums*, *Macrovipera lebetina*, *Cryptelytrops erythrurus*, *Daboia russelii*, *Protobothrops jerdonii*, *Echis carinatus, Hypnale hypnale* [51]. Each of these venoms must be obtained from male, female, young, and adult specimens, and from various locations/countries, to cover the venom’s geographical variation. Furthermore, in the case of elapid venoms, toxin fractions containing only the lethal neurotoxins, and devoid of high-MW immunogenic non-toxic proteins, can be used [12]; this could further increase the number of immunizing elapid venoms, e.g., WHO Category 2 snakes and sea snakes, to widen the paraspecificity of the syndromic neurotoxic polyvalent AV [15].

It should be mentioned that, as well as the wide paraspecificity of the s-pAVs, and the removal of the need to identify culprit snakes, the potency of the s-pAVs, at least with regard to the ‘neuro s-pAV’, should be higher than that of the corresponding ‘combined pAV’ discussed in the form of the anti-elapid-plus-anti-viperid AV (Section 2.5). Furthermore, the s-pAVs have been shown to exhibit immunochemical and biochemical properties comparable to those of the corresponding monovalent antivenoms prepared under similar conditions [52].

### 3.3. Future Prospects for Producing ‘Universal s-pAVs’

It has been hypothesized that, when a horse is exposed to a ‘diverse toxin repertoire’, the animal should respond by producing a widely paraspecific antiserum against the toxins. Thus, it has recently been shown that, through the use of the toxin fractions of 12 Asian elapids as immunogens, a ‘universal’ antivenom against at least 36 elapid snakes of 28 species in 10 genera, inhabiting over 20 countries on four continents, was prepared [15]. It is expected that, with the ‘diverse toxin repertoire’ immunization strategy [13], a similar achievement could be demonstrated for ‘universal’ s-pAVs against neurotoxic venoms.

With regard to the ‘universal’ s-pAV against hematotoxic viper venoms, a similar strategy may be applicable. A possible approach is to use affinity chromatography to purify the major toxic enzymes of various medically important viperid venoms, and use them as immunizing antigens. For example, affinity matrixes can be prepared for the purification of snake venom serine proteases [53,54], for phospholipase A_2_ [55,56], and for hemorrhagic metalloproteases [57,58]. After a dozen or so medically important viper venoms have been sequentially passed through such affinity columns, the purified enzymes from each of these venoms can be eluted from the columns, and collected. A mixture of 2–3 µg of each of these enzymes from each venom can then be used as the primary immunogen, to generate a widely paraspecific ‘hemato s-pAV’. With the previously observed wide cross-reactivity of the antisera generated against these enzymes [48,49], it is likely that the antisera generated from the above ‘diverse toxin repertoire’ immunization [13] could result in a ‘universal hemato s-pAV’.

If this experiment were to be successfully carried out, then the two ‘universal’ syndromic antivenoms should be able to cover the few dozen countries of sub-Saharan Africa, and possibly South Asia. With the economy of scale, it should be possible to produce these ‘universal’ s-pAVs in large volumes at a reduced cost, and allow them to be affordable in various poverty-stricken areas of the world. If realized, these ‘universal’ s-pAVs would greatly reduce the fatalities caused by snakebite envenoming.

## 4. Conclusions

Snakebite envenoming causes a high rate of morbidity and mortality in developing countries. Although specific plasma-derived antivenoms have been used for about a century, they are not all effective, and are expensive and in short supply. Polyvalent antivenoms have the advantage of neutralizing many different snake venoms from a wide region. In addition, they remove the need to identify the culprit snake (a requirement in selecting a specific monovalent antivenom), and they are cheaper to produce. Different procedures have been used to produce polyvalent antivenoms, with ensuing advantages and drawbacks. Syndromic polyvalent antivenoms constitute a relatively new therapeutic modality, with the potential to alleviate snake envenoming problems. The syndromic polyvalent antivenoms described here are in line with the first WHO public-benefit Target Product Profiles for snakebite antivenoms for sub-Saharan Africa [59]. While novel therapeutic alternatives based on recombinant antibody technologies are being rigorously pursued, plasma-derived polyvalent antivenoms, especially syndromic polyvalent antivenoms, are already available as lifesaving therapeutics.

## Figures and Tables

**Figure 1 toxins-15-00517-f001:**
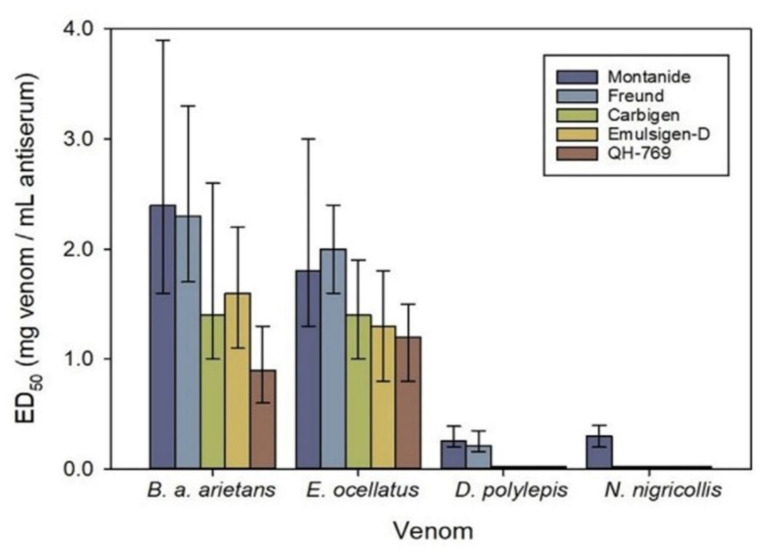
The horse antibody response to immunization with viperid (*Bitis arietans arietans*, *Echis ocellatus*) and elapid (*Dendroaspis polylepis*, *Naja nigricollis*) venoms, using different immunological adjuvants. The antibody response is expressed as mg venom neutralized per ml of antiserum. This Figure was published in Arguedas et al. ‘Comparison of adjuvant emulsions for their safety and ability to enhance the antibody response in horses immunized with African snake venoms’. *Vaccine X*
**2022**, *12*, 100233 [23].

**Figure 2 toxins-15-00517-f002:**
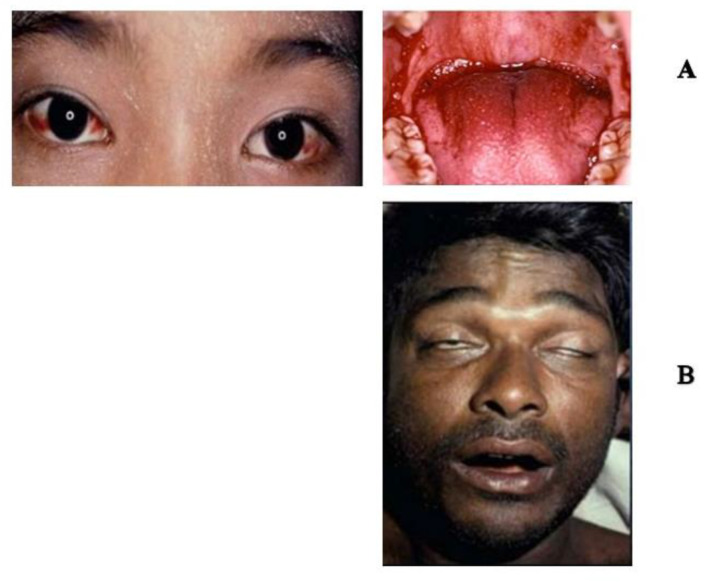
(**A**) Patients with coagulopathy and bleeding (signs of hematologic disorder) caused by a *Daboia russelli* bite. (**B**) A patient showing bilateral ptosis (signs of neurotoxic poisoning caused by an elapid (*Bungarus caeruleus)* bite. Photographs courtesy of Professor David A Warrell.

**Figure 3 toxins-15-00517-f003:**
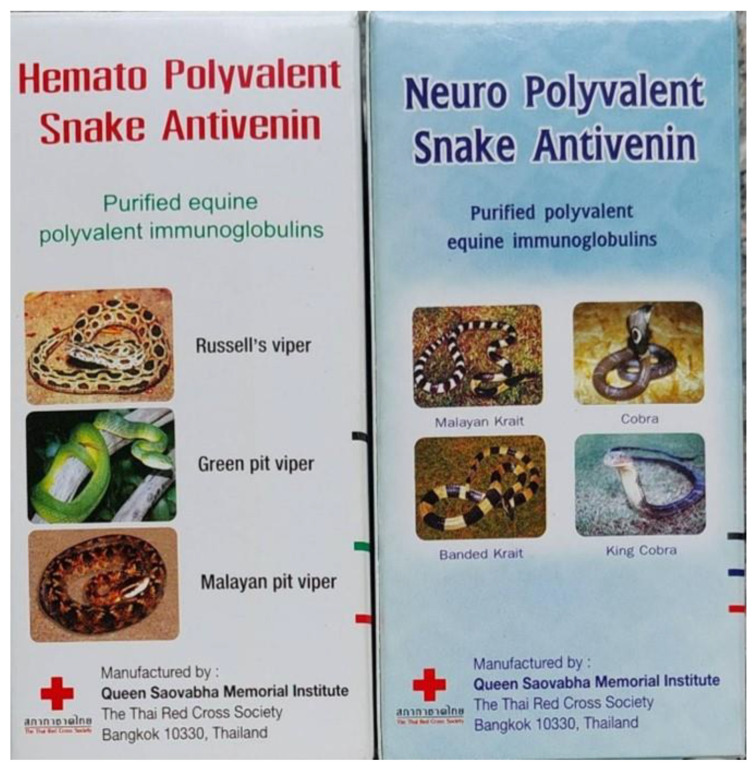
The two syndromic polyvalent antivenoms: “Neuro Polyvalent Antivenin” (**right**) and “Hemato Polyvalent Antivenin” (**left**), produced by Queen Saovabha Memorial Institute, The Thai Red Cross Society.

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
