# Peer review of "Polyvalent Snake Antivenoms: Production Strategy and Their Therapeutic Benefits"

_toxins, 2023, doi:10.3390/toxins15090517_

Round 1

Reviewer 1 Report

Comments for toxins-2508641:

In this paper, authors described review article of polyvalent antivenoms (pAV) against snake-bite.  This review article is well written with recent references, however, I have a few concerns and queries that may be considered before the publication of the article.

1)     The authors include some photographs showing sumptoms of venomous snake-bite patients as Figure 2. However, no description about Figure 2 in the text. Detailed descriptions about Figure 2 are required. What are differences between left and right panels in A and B, respectively.  What kind of snakes were contributed for viperid bite (A) and elapid bite (B). By the way, do authors have permission from the patients as well as the photo provider to publish the photos?

2)     In the conclusion section, authors describe ‘The syndromic polyvalent antivenoms described here are in line with the first WHO public-benefit Target Product Profiles (TPPs, July 2023) for snakebite antivenoms for sub-Sahara Africa.’  Reference about TPPs, July 2023 should be probided.

Minor editing of English language required.

For what reason do authors use upper and lower case letters for 'hematotoxic s-pAV' , ' ‘non-neurotoxic’ Hematotoxic s-pAVs' etc.?

Author Response

Response to Reviewer 1

Reviewer #1

In this paper, authors described review article of polyvalent antivenoms (pAV) against snake-bite. This review article is well written with recent references; however, I have a few concerns and queries that may be considered before the publication of the article.

  • The authors include some photographs showing symptoms of venomous snake-bite patients as Figure 2. However, no description about Figure 2 in the text. Detailed descriptions about Figure 2 are required. What are differences between left and right panels in A and B, respectively. What kind of snakes were contributed for viperid bite (A) and elapid bite (B). By the way, do authors have permission from the patients as well as the photo provider to publish the photos?

Author’s response: I have added the details on Figure 2:

Figure 2. A. Patients with coagulopathy and bleeding (signs of hematologic disorder) caused by viperid (Daboia russelli) bite. B. A patient showing bilateral ptosis (signs of neurotoxic poisoning) caused by elapid (Bungarus caeruleus) bite.

Photos by courtesy of Professor David A Warrell.

These photos were kindly provided by Professor David A Warrell and have been previously published, so I believe that they had the permissions of the patients.

Thank you.

  • In the conclusion section, authors describe ‘The syndromic polyvalent antivenoms described here are in line with the first WHO public-benefit Target Product Profiles (TPPs, July 2023) for snakebite antivenoms for sub-Sahara Africa.’ Reference about TPPs, July 2023 should be provided.

Author’s response: I have added the reference #59 for the WHO TPP.

Comments on the Quality of English Language Minor editing of English language required.

  • For what reason do authors use upper and lower case letters for 'hematotoxic s-pAV' , ' ‘non-neurotoxic’ Hematotoxic s-pAVs' etc.?

         Author’s response: I have corrected all these terms in the manuscript using lower  

            case letters. Thank you very much.

Reviewer 2 Report

The manuscript reviews various polyvalent antivenoms (pAVs) and their therapeutic benefits. The multiple parameters involved in producing pAVs and their characteristics provide an optimistic view regarding the use of pAVs in treating snakebites. The relevance of this work aligns well with WHO policies aimed at combating snakebite envenomings on a global scale. This manuscript holds utmost significance and has the potential to become a cornerstone in the efforts of groups dedicated to enhancing snakebite envenomation treatments. The authors need to address the following points:

1.     The paper would be much more interesting if the authors provide more information on the clinical efficacy of syndromic polyvalent antivenoms (e.g. PMID: 34876815) in this manuscript.

2.     In several sentences, the reference is missing. For example:

-        Lines 76-77 “….immunization with a mixture of more than 5-6 venoms may result in a total protein antigen loads that can overwhelm the immunized animal and resulting in a low titer of the produced antibodies.” A reference should be added to support it.

-        Line 196 “…in Thailand, O. hannah (King cobra) is WHO Category 2 snake and although it causes very few incidents of envenomation,….”. A reference should be added to support it.  

3.     The authors should replace “snake-bite” with “snake bite”.

4.     Line 114: the authors should delete “4” from “sub-4Sahara”.

5.     Lines 21, 126, and 128: Please use lower case for “syndromic pAVs”; Line 142: “three finger toxins”

6.     Lines 149-150, 195, 229, 232, and 233: The authors should spell the snake genus for first-time use.

7.     Line 241: The authors should use lowercase for the species name “nigricollis”.

8.     “paraspecificity” or “para-specificity”?

The manuscript needs careful editing and revision with respect to English grammar, sentence structure, and wording. Some corrections have been identified, but not all have been listed in this comment. Please check carefully if revision occurs.

Author Response

Response to Reviewer 2

Reviewer 2

The manuscript reviews various polyvalent antivenoms (pAVs) and their therapeutic benefits. The multiple parameters involved in producing pAVs and their characteristics provide an optimistic view regarding the use of pAVs in treating snakebites. The relevance of this work aligns well with WHO policies aimed at combating snakebite envenomings on a global scale. This manuscript holds utmost significance and has the potential to become a cornerstone in the efforts of groups dedicated to enhancing snakebite envenomation treatments.

The authors need to address the following points:

  1. The paper would be much more interesting if the authors provide more information on the clinical efficacy of syndromic polyvalent antivenoms (e.g. PMID: 34876815) in this manuscript.

Author’s response:  I very much agree with the Reviewer that inclusion of clinical aspect on the syndromic polyvalent AVs would make the manuscript more interesting. However, with the author’s limited background knowledge on clinical studies and the limited time available for submission of this special issue, I regret that I could not include this very important aspect of pAVs.

  1. In several sentences, the reference is missing. For example:

         - Lines 76-77 “….immunization with a mixture of more than 5-6 venoms may result in a total protein antigen loads that can overwhelm the immunized animal and resulting in a low titer of the produced antibodies.” A reference should be added to support it.

Author’s response: Various studies on polyvalent AV production using many venoms inevitably encounter lower antibody response against some immunizing venoms. For example, the study by Arguedas et al., [23] (Figure 1 of this manuscript) showed rather low neutralizing antibody even only 4 venoms were used although they were mixture of elapid and viperid venoms. The information obtained from various AV producers often indicated lower antibody titers against some immunizing venoms. The immunization procedure may contribute to this observation, but as the number of immunizing venoms increases, the antibody titers against some venoms would decrease. Regrettably, direct quantitative comparison on the antibody responses as a function of number of immunizing venoms, as far as I am aware of, has not been carried out.

- Line 196 “…in Thailand, O. hannah (King cobra) is WHO Category 2 snake and although it causes very few incidents of envenomation,….”. A reference should be added to support it.

Author’s response: I have included the data on this point with the reference (#4) to it (Lines 201-203)

  • The authors should replace “snake-bite” with “snake bite”.

Author’s response: corrected, thank you.

  1. Line 114: the authors should delete “4” from “sub4Sahara”.

Author’s response: corrected, thank you.

  1. Lines 21, 126, and 128: Please use lower case for “syndromic pAVs”;

           Line 142: “three finger toxins”

Author’s response: I have corrected this error though out the ms, thank you.

  1. Lines 149-150, 195, 229, 232, and 233: The authors should spell the snake genus for first-time use.

Author’s response: corrected, thank you.

  1. Line 241: The authors should use lower case for the species name “nigricollis”.

Author’s response: corrected, thank you.

  1. “paraspecificity” or “para-specificity”?

            Author’s response: “paraspecificity” is now used in the manuscript.

  1. Comments on the Quality of English Language The manuscript needs careful editing and revision with respect to English grammar, sentence structure, and wording. Some corrections have been identified, but not all have been listed in this comment. Please check carefully if revision occurs.

Author’s response:  Thank you very much. I have asked my colleagues, one American and one Canadian to edit the manuscript and I hope it is now in better form.